# Design of a micro-learning framework and mobile application using design-based research

Heydy Robles[1], Miguel Jimeno[2], Karen Villalba[1], Ivan Mardini[3], César Viloria-Nuñez[3] and Wendy Florian[4]

[1] Language Department, Universidad del Norte, Barranquilla, Altántico, Colombia
[2] Systems Engineering Department, Universidad del Norte, Barranquilla, Atlantico, Colombia
[3] Electrical Engineering Department, Universidad del Norte, Barranquilla, Atlantico, Colombia
[4] Graphic Design Department, Universidad del Norte, Barranquilla, Atlantico, Colombia



Corresponding author
Miguel Jimeno,
majimeno@uninorte.edu.co

## ABSTRACT

Traditional learning techniques have evolved slowly and have yet to adapt the course content delivery to today's students' approaches to acquiring new knowledge. However, micro-learning has become popular in e-Learning environments as a course design technique due to short attention spans, demand for small chunks of information, and time constraints. Hence, it has been selected for creating reading mobile applications provided to the nature of its learning approach. In order to describe the multiple iterations of design, development, and evaluation of this general framework, a methodology named Design-Based Research (DBR) is implemented. First, the article presents the abstract framework components and a cloud-based software architecture that allows a modular approach to creating such applications. The pathway developed through adapting the iPAC framework, which involves personalization, authenticity, and collaboration, is part of the methodology used to design the app under pedagogical and technological considerations. The process demanded the following phases: analysis and exploration, design and construction, evaluation and reflection, redesign and reconstruction, and final critical reflections. Four applied instruments also validate the framework implementation: The iPAC Rubric, an aphorisms checklist, a pre and post-test, a focus group, and a usability test taken by 28 students in a private university in Colombia. Findings indicated that Design-Based Research (DBR) methodology emerged as an appropriate tool to encounter the needs behind reading applications design due to its sequence of operations yields results successively closer to adequate usability standards and smooth implementation. They also reveal the positive impact of new types of texts on students' motivation and awareness toward other reading strategies and micro-learning. This impact indeed proved the proposed framework's effectiveness for designing micro-learning applications.

# INTRODUCTION

E-Learning continues to grow worldwide, thanks to the effectiveness of e-Learning systems. In detail, there has been an exponential increment of content from online learning platforms (including feedback and recommendations) on learning platforms supported by

traditional universities and companies (*Kolakowski & Ebrahim, 2021*; *Tan & Shao, 2015*). This type of learning's constant growth creates multiple online courses, diplomas, and programs of various lengths, but this has not guaranteed student engagement success rates. According to reports from different universities and countries, traditional e-Learning systems have failed to secure student engagement, thus generating dropouts that vary broadly, ranging between 20% and 80% (*Rostaminezhad, Porshafei & Ahamdi, 2019*; *Sun et al., 2008*). The studies found that student factors (including academic background and skills) accounted for 55% of the reasons for failure, while course or program factors (including course design) accounted for 20% of the reasons. Other factors included environmental factors. Other researchers categorize the factors that influence the high dropout rates into (1) factors related to the learner and its context and (2) aspects concerning the course design (*Martínez, Leite & Monteiro, 2016*). While most research focuses on the student's factors, the focus has been less on the factors associated with how the content providers deliver the courses to them. It is essential to highlight that revised studies disagree with similar dropout rates.

The research literature has evidence on the course design factors influencing the effectiveness of e-Learning systems. In detail, feedback and recommendations include providing content relevant to students' interests and experience, allowing self-direction and self-exploration from students, and encouraging student participation with interactive and exciting content (*Martínez, Leite & Monteiro, 2016*). Many e-Learning systems use the course design approach inherited from traditional educational methods, although there is a recent approach to designing adaptive e-Learning systems (*Bradac & Walek, 2017*; *Kulaglić et al., 2013*). However, new pedagogical methodologies must guarantee that students will engage and participate with their peers in the learning process. With the arrival and explosion of information generated by social networking tools, traditional educational structures find it more challenging to achieve the initially set learning objectives due to the supposed adverse effects of social networking on learning (*Akram & Kumar, 2017*; *Rostaminezhad, Porshafei & Ahamdi, 2019*).

Micro-learning has emerged as a learning approach in which information is processed in small manageable pieces to enable better retention, engage students and obtain micro-content for flexible learning environments such as mobile learning and mobile applications. Mobile applications are part of the daily culture and the trend of the digital learning field (*Nami, 2020*). Applications are specialized programs to run on mobile media platforms (*Castek & Beach, 2013*). They are easily available and accessible alternatives to classroom learning and serve social influence, novelty, engagement and activity (*Menon, 2022*). They compel affordances that are cues of an artifact's potential uses by an agent in each environment and refer to the agent's possibilities for action (*Burlamaqui & Dong, 2014*). The term 'affordance' was conceived by *Gibson (1977)* as a quality of a particular object that permits it to act. Since applications rely on devices, it is necessary to analyze their affordances, such as ubiquity, engagement, portability, and reachability (*Beach & O'Brien, 2014*), to understand how the micro-learning process aids students in circumventing the sensation of mental exhaustion. Self-assessment and rehearsing of the

material create stronger neural network connections within the brain, convey short-term to long-term memory, and improve previous performance.

This article introduces a fresh perspective on the factors affecting students' satisfaction with micro-learning. From there, it offers a cutting-edge modular architecture for creating mobile apps that incorporate micro-learning in an e-Learning setting. It emphasizes technology and course design aspects to impact students, increase engagement, and reduce dropout rates. The design of the framework and resulting application uses Design-Based Research (DBR). The developed application is tested with a group of students, achieving a measurable positive outcome in the English learning process. Therefore, the contributions of this article are as follows:

1. The first use of micro texts in mobile environments to develop English reading skills by using collaborative approaches to encourage students' participation.
2. A new software-based modular framework that defines functional components to help designers of mobile learning applications to implement parts as needed.
3. The blend of the DBR methodology and the iPAC framework contributed to innovative applications that consider students' needs and interests in their learning process.

The remainder of the article is then organized as follows:

1. First, a background and related work is exposed to establish the terminology and past work.
2. The micro-learning framework is explained in more detail in the next section, with details of the conceptual components and its cloud-based and mobile implementation.
3. The next section designs the experimental setup, followed by the results of the experiments.
4. The article concludes with a discussion, conclusions and expected future work.

## BACKGROUND AND RELATED WORK

### Micro-learning

Micro-learning is a technique where people learn with small chunks of information during short periods (*Hug, 2005*; *Hug, 2007*; *Jomah et al., 2016*). It has increased its popularity for course designs in e-Learning environments. Using micro-learning has three goals: By breaking up the knowledge into manageable chunks, you can: cut down on the number of information students need to access; redefine the learning process and the learning environment; and encourage students to personalize their learning (*Mohammed, Wakil & Nawroly, 2018*; *Trowbridge, Waterbury & Sudbury, 2017*). This study plans to use micro-learning to adapt the course design to the students' current profiles. It also aims at applying the micro-learning objectives defined in the literature.

Recent statistics in Colombia have revealed significant weaknesses in reading in both languages. Only 1% of students achieved the highest levels on levels 5 and 6 in the Programme for International Student Assessment (PISA) reading test. Besides that, 50% can define the main idea in a moderate-length document and find information based on

explicit and straightforward criteria. Students can focus on the text's intent and type (*OECD, 2019*) when individually guided. In response to these needs, a framework for developing mobile applications that employ micro-learning methodologies will be developed, which will be applied to the development of an application for improving reading comprehension. Researchers have recently adapted new e-Learning methodologies using micro-learning for new users. Information is most frequently accessed digitally, visualized, and delivered in short chunks to new students (*Zhang & West, 2020*). Research has found that students can remember content learned using micro-learning strategies longer than traditional approaches (*Mohammed, Wakil & Nawroly, 2018*). Micro-learning is a way of performing Competency-Based Education (CBE) (*Zhang & West, 2020*). While traditional education relies on teacher-led training, CBE focuses on individual learners' goals. This type of learning allows students to select which skills they would like to improve and choose an option in an application that would help them achieve their goals. Many studies have previously covered CBE, and one of them states the key advantages and disadvantages of it (*Barman & Konwar, 2011*). The advantages are many, which makes it attractive to educators. For example, the CBE process is learner-centred, as educators aim to guarantee that students achieve the expected competencies. Achieving the competencies help improve the teacher-learner relationship as students might appreciate their renewed focus on their learning process and not the traditional course objectives. On the other side, the authors state that higher education institutions that implement it must deal with existing national-level regulations that might prevent them from changing education paradigms. Definitions of competencies should also be standardized across similar knowledge areas. For second language instruction, micro-learning has offered benefits and has shown that these students enjoyed studying more, were more actively involved in the learning process, and were more conscious of the learning process as a whole. This led to the growth of their autonomy and self-regulated learning (*Hosseini, Ejtehadi & Hosseini, 2020*; *Khong & Kabilan, 2022*).

Previous work, such as the one presented in *Emerson & Berge (2018)*, has shown how companies have used flexible micro-learning strategies and engaged employees to use mobile devices to participate. According to previous benchmarks, those authors stated that companies had preferred on-demand learning and access to updated information when needed. The proposal of this article satisfies this requirement since it could quickly adapt to different educational environments. The authors of *Bruck, Motiwalla & Foerster (2012)* proposed a framework for evaluating multiple aspects of micro-learning applications on the evaluation side. The purpose was to establish different evaluation criteria to determine the real impact of such applications. Criteria included micro-interactions, levels of personalization, and learning improvement. Another aspect outlined by other authors who have explored micro-learning is its positive impact on self-regulation components such as forethought, performance and reflection. This approach strengthened students' fundamental psychological requirements for self-perceived competence, relatedness, and autonomy and enhanced their exam results in terms of factual knowledge, which is crucial for the needs Colombian students have (*Shail, 2019*; *Shamir-Inbal & Blau, 2022*). Given the serious reading deficiencies noted in Colombia, a framework for creating mobile

**Table 1 Usability models and its set of attributes.**

| Usability models | A | B | C | D | E | F | G | H | I | J | K | L |
|---|---|---|---|---|---|---|---|---|---|---|---|---|
| Shackel | X | | X | | | X | | | | | | |
| Nielsen | | X | X | | | X | | | | | | X |
| Abran | X | X | X | | | X | | | | | | |
| Seffah | X | | X | X | X | X | | X | | | | X |
| Dubey | X | X | X | | | X | | | | | | |
| Schinder-man | X | X | X | | | X | | | | | | |
| Preece | | X | X | | | X | | | | | | |
| Gupta | X | X | X | X | X | | | | | | | |
| ISO 25010 model | X | X | X | | | X | X | X | X | X | X | |

applications that use micro-learning techniques will be created. It is unclear whether using aphorisms as a course design strategy is appropriate given the lack of clarity on the impact on students' satisfaction in terms of usability, enhancement of inference making, engagement with micro texts, and pedagogical aspects.

## Usability

Mobile learning's three pedagogical features (personalization, authenticity, and collaboration) are distinguished from the sociocultural theory perspective. These are a part of the iPAC framework, which gives students autonomy over their learning. Furthermore, it offers contextualized tasks in group-based networked environments (*Bano, Zowghi & Kearney, 2017*). There are also planning imagination and creativity-related instructional advantages. These are the cornerstone for usability assessment and developing learning objectives (*Burden & Atkinson, 2008*; *Vincent, 2012*). The affordances that emerge from evaluating applications are relevance, customization, feedback, and usability. This final affordance is the focus of one of the research questions in this article.

ISO 9126 is part of the ISO 9000 standard, the most recognized standard for quality assurance that defines a set of quality attributes. Here, usability appears through the following characteristics: understandability, learnability, and operability (*Spriestersbach & Springer, 2004*). Similarly, it is treated as an affordance that allows students to launch and operate an application (*Beach & O'Brien, 2014*) independently. Therefore, ISO 9126 served as a reference to define usability and understand the concept. For the selection of the app attributes, different usability models from the last three decades have been exposed as contributing to developing usable software systems, as shown in Table 1. This study aimed to collect all the models and create one single usability assessment indicators table. The included attributes are 14: (A) effectiveness, (B) efficiency, (C) satisfaction, (D) productivity, (E) universality, (F) learnability, (G) appropriateness, (H) recognizability, (I) accessibility, (J) operability, (K) aesthetics, and (L) error protection. Effectiveness is the ratio of tasks, the correct ones, and the frequency of errors, while efficiency measures the time required to complete the job.

In the same way, satisfaction represents the trust, comfort, pleasure, and usefulness of the application. Productivity measures the cost-effectiveness of performing a task by the

**Table 2 Application usability assessment indicators.**

| Usability attributes | Indicator aspect | Indicator category |
|---|---|---|
| H | Accessibility | Personal preferences, instantaneous support, usage approach |
| J | Aesthetics | Color and icons, interface comfort |
|  | Assessment strategy | Degree of preparation, assessment performance, consistent objectives |
| B | Completeness | Indicating links, browsing interface, overall structure |
| I | Consistency and functionality | Similar formats, clear functions, convenient interface |
| D | Convenience | Loading speed, personalized environment |
| G | Course management | Links to information, personalized resources, ease of uploads, download and views |
| A | Error prevention | Multiple operations, cancellation of function, hints and warnings |
| F | Intention to use | Continued use, emotional improvement, recommend to others |
| E | Interactivity feedback and help | Improved communication, regular feedback, keeping track of progress |
| F | Memorability | Question help, options program, hint windows, Clear interface |
| C | Perceived usefulness | Grades improvement, learning efficacy, improved knowledge, facilitating learning |
| K | Reducing redundancy | Amending the errors, flexible interface, reading materials |
| C | User satisfaction | Information use, exceeding of expectations, expected results |
| I | Visibility | Reasonable arrangement, clear functions, effective layout |

user and the minimum number of actions needed. Universality represents compliance with international standards and other cultures' backgrounds. Learnability measures how easy to remember are system's functionalities and clarifies error occurrence and learning time.

Finally, aesthetics determine the system's attractiveness and convenient customization, and the error protection category measures error avoidance, recoverability, and validity checking (Hasan & Al-Sarayreh, 2015). If this affordance is evaluated, it guides the design issues that might emerge from the user perspective. Therefore, designers must implement a valid assessment indicator based on the Technology Acceptance Model (TAM) for perceived usefulness, ease of use, and behavior intention, among other subcategories (Jou et al., 2016).

Table 2 shows the current usability assessment indicators for applications. Its origin is directly related to the usability attributes of the usability models shown in Table 1. Each indicator aspect is validated and coherent with the previous usability models. A more detailed set of categories offers a holistic quality approach for evaluating mobile applications' usability and user experience. It maintains the same usability issues from the past, but now it responds to the challenges and complexities of new applications. This study measured the usability of the current framework by applying two usability tests to students at two different stages of the design. In here, the indicators are explicitly shown to react to the student's perception of the mobile application's usability.

## Frameworks for e-Learning creation
The work presented in Leal, Queirós & Ferreira (2010) proposed a survey of e-Learning frameworks. The article states that general or abstract frameworks define work environments designed to solve problems in different domains. A software framework usually establishes a set of tools developers use, typically including support programs,

runtime environments, and libraries. These frameworks help developers decrease the need for new code by leveraging the development and increasing dependability. The IEEE Learning Technology Standards Committee (LTSC) (*IEEE, 2003*) proposed the IEEE Learning Technology Systems Architecture as a standard, or abstract framework, for technology-based learning. The proposed architecture has Learner, Evaluation, Delivery, and System Coach components. The architecture is supported by data management components such as learning resources and a records database. Previous work has also proposed frameworks to find a standard structure to fit various e-Learning course design requirements. One example is the work from *Aljohani et al. (2019)*, whose proposal for tracking student analytics could adapt to any course structure. Another example is presented in *Rauschnabel, Rossmann & Tom Dieck (2017)*, which explains the drivers of attitudinal and intentional reactions in players of augmented reality games. That framework focused on establishing connections between factors and elements under study. The previous work on frameworks shows proposals that serve standard features found in e-Learning environments. However, there is a need for a framework that helps developers build applications that implement e-Learning components and take into account factors that could encourage students' participation and improve their engagement with the tool.

## Reading comprehension applications

Today reading is a complex and multifaceted process whose nature changes with experience and development. It involves cognitive skills and various word-level and text-level reading skills (*Geva & Ramírez, 2016*; *Jamshidifarsani et al., 2019*). Students who develop strong core reading skills also build comprehension skills, thus the importance of developing reading comprehension from an early age. The prevailing view is that reading is an interactive process involving both bits of knowledge of the world and the language. They coordinate and interact to contribute to the text's comprehension (*Williams & Moran, 1989*). There are three levels of reading comprehension: the literal, which corresponds to reading the lines; the inferential or reading between the lines; and the critical level, which corresponds to reading behind the lines (*Basaraba et al., 2013*). In this new digital era, the education process slowly leaves traditional paper reading behind, and students are now learning with the latest technological trends (*Liu, 2019*). There has been increased interest in mobile technologies' potential to support reading comprehension and motivate students (*Cheung & Slavin, 2013*; *Gómez-Díaz, García Rodríguez & Cordón-García, 2015*; *Hashim & Vongkulluksn, 2018*).

Conversely, since reading does not get enough attention in primary and secondary education, university students must develop abilities to make their academic job easier. However, it is also believed that by the time pupils get to college, they should have mastered this ability. At the same time, it is assumed that students should already have developed this skill by reaching university (*Bosley, 2008*). Reading comprehension enhances academic, professional, and civic performance, claims *Paul & Clarke (2016)*. Researchers conducted in Latin American settings have found that university students lack the comprehension skills required to comprehend a text and cannot organize their information into summaries or distinct texts (*García, Nájera & Téllez, 2014*). Besides,

many reading comprehension applications aim at students who wish to learn a second language, and only a few applications with Spanish as the mother language to improve reading comprehension (*León, Bravo & Fernández, 2017*). Similarly, the research work led by *Roncal Mejia (2015)* established that there are few Spanish applications designed for reading comprehension despite there being an emerging necessity for this. Additionally, the Spanish applications cover children's and very young learners' needs. Few options are offered to young adults and adults facing reading comprehension challenges (*Vásquez et al., 2019*).

Previous work has proposed frameworks to find a standard structure to fit various e-Learning course design requirements. One example is the work from *Aljohani et al. (2019)*, whose proposal for tracking student analytics could adapt to any course structure. Another example is presented in *Rauschnabel, Rossmann & Tom Dieck (2017)*, which explains the drivers of attitudinal and intentional reactions in players of augmented reality games. That framework focused on establishing connections between factors and elements under study.

## Research directions

In a previous systematic evaluation of literacy learning and reading comprehension applications, the findings reveal that current proposals for use by language learners are generally weak in many of the salient features and affordances of mobile learning and training (*Gutiérrez-Colón, Frumuselu & Curell, 2020*; *Israelson, 2015*). They are particularly so in the case of collaboration, which is recognized as both a critical affordance of mobile learning and an essential skill that underpins effective language learning and reading comprehension. Reading comprehension applications tend to be a 'one size fits all' learning experience and omit the opportunities to make reading comprehension more meaningful, engaging, and realistic.

The lack of opportunities for collaboration suggests that designers and creators of pedagogical applications are more focused on traditional learning approaches. Information is 'delivered' to the student rather than a sociocultural model. The learning is deemed more participative, social, and mediated through technologies such as the mobile device itself. The findings from this study indicate that applications need to be designed to exploit more opportunities for collaboration between learners and avoid artificial or created texts for educational purposes exclusively.

The following research questions were proposed in light of what the background work shows us of what is missing in the literature.

1. How should a framework for a reading comprehension micro-learning mobile application be described?
2. How did the students perceive the usability of the mobile application?
3. How did the students perform after using the reading comprehension micro-learning mobile application?
4. Is a combination of DBR and iPAC suitable to design and develop modular mobile applications for improving English and Spanish reading skills with innovative educational practices?

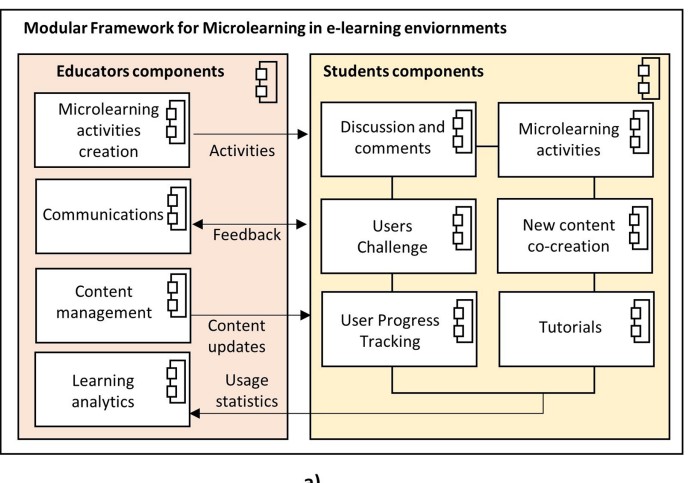

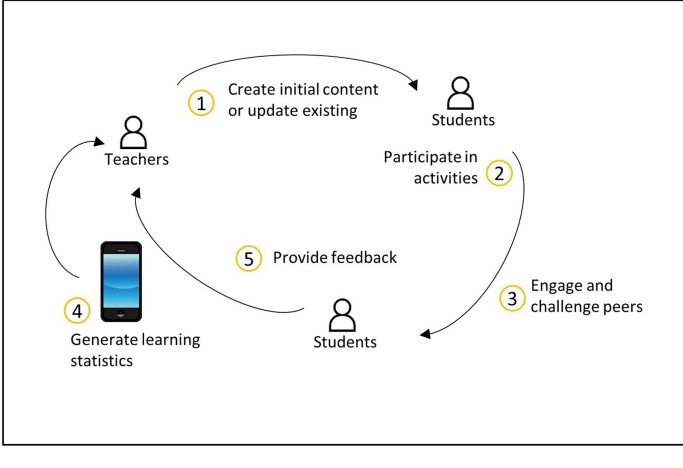

**Figure 1** **Framework for microlearning in an e-Learning environment.** (A) Conceptual framework and (B) interaction between components.

## DESIGN OF THE MICRO-LEARNING FRAMEWORK

The framework proposed in this article is considered a combination of an abstract framework and a software framework, as previously defined in the Background section. An abstract framework defines abstract components and their interaction. On the other side, a software framework defines tools to simplify the development process. This section then explains the framework in two parts. First, it shows the abstract components and their interaction, and then the proposed cloud-based architecture defines the software side of the framework.

### Framework components and their interactions

Figure 1 shows two views. Figure 1A part shows an abstract description of the framework components. This view details the two sides of a complete design of a mobile e-Learning project based on a micro-learning architecture. It is important to note that the components' design allows the implementation to be modular. The components that should be included in the mobile application are displayed on the students' side. They build on gamification, which has been shown to be beneficial, to promote engagement and adherence to the program (*Caro-Alvaro et al., 2017*). Some strategies include reward systems, levels, and immediate feedback such as success messages when challenges are accomplished (*Atkins, Wanick & Wills, 2017*). The components of this side of the architecture in Fig. 1A are the following:

- Micro-learning activities: this is the business activities core of the application developers will design using this framework. Activities may include short videos, readings, or any concise bite-sized information the student can learn (*Bothe et al., 2019*).
- Players Challenge: the gamification process is shown in this component by using the leader boards strategy or point systems (*Atkins, Wanick & Wills, 2017*).

- Discussion: this component aims to engage interaction between students. Previous research, such as the one presented in *Saadé & Huang (2009)*, has proven the impact on learning processes when students are encouraged to participate in online forums.
- New content co-creation: this component aims to encourage students to create their content to share as a micro lesson. Previous work has also shown the importance of allowing students to develop and design content, as shown in *Nahar & Cross (2020)*, who proposed a Student-Staff Partnership (SSP) to encourage students to participate in the process.
- User progress: it allows the students to see their progress in the points system and provide feedback about their performance.
- Tutorials: this component provides users with digital content for help without interacting with educators.

The educators' side in Fig. 1A describes the components of the e-Learning project's server-side. The components are the following:

- Learning analytics gathers information and establishes connections with each student's profile (*Hwang, Chu & Yin, 2018*).
- Micro-learning activities creation: educators use this component to feed the micro-learning section with learning content.
- Communications: this component establishes a bidirectional communication channel with the students inside the platform.
- Content management: it updates any other information needed.

Figure 1B shows the interaction between the components shown in Fig. 1A. The flow of the information starts with the teacher, which assigns or creates the original activities, then the student completes them, and some of them will even challenge their peers. Finally, the feedback comes back to the teacher, which might redefine or create new content thanks to the information received.

## Cloud-based software framework architecture

The proposed framework architecture implementation describes the different layers in Fig. 2. It assumes the implementation lies in a cloud. The layers are the following:

- Cloud computing layer: the cloud provides different advantages, and it is the leading hosting architecture used nowadays for e-Learning information systems to manage requests across the Internet (*Bosamia & Patel, 2016*; *Sana & Li, 2021*).
- Data processing services layer: it contains three primary data processing services to allow scalability depending on the demand for each service.
- Back-end API layer: This layer uses micro-services software architecture, which previous authors have shown to help e-Learning environments (*Kapembe & Quenum, 2018*).
- Data-Analytic services layer: it allows designers to build their own data analytic services or select one of the many services available.

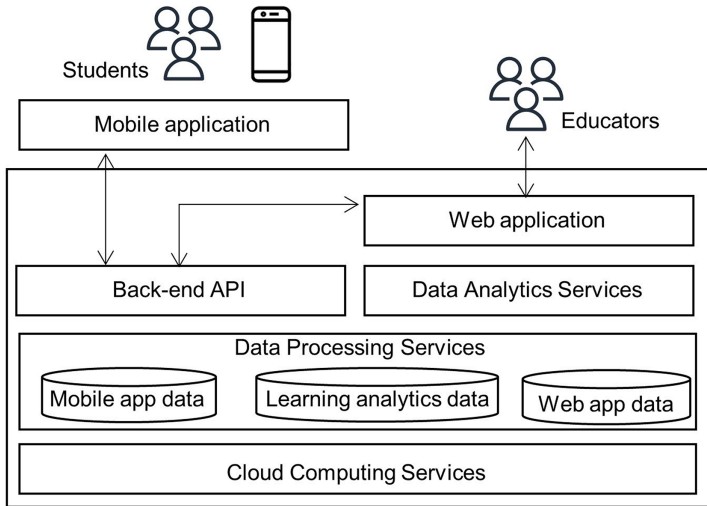

**Cloud-based architecture for Framework implementation**

**Figure 2** Cloud-based architecture for framework implementation.

The interaction of the components in the previous figure is as follows. Cloud computing services provide hosting functionalities, including virtual machines or containers, load balancing, database server collocation, networking, and security. On top of this layer sits the data processing services, exposed as microservices, to offer an interface to the database deployment.

Mobile applications come on top of the other layers for students and educators and consume the services exposed in each layer. While the mobile application connects to the back-end API, the web application for the educators serves as a front-end tool for them to find reporting tools. The Data-Analytic component generates compiled information regarding students' performance according to multiple variables.

## Mobile application design and implementation

The components of the mobile application consist of two blocks: Front-End and Back-End. The descriptions are below.

### Front-end

The Front-End application uses React Native for IOS and Android, a JavaScript framework that facilitates the programming phase of the implementation. Figure 3 shows the screenshots of the Front-End. Figure 3A offers the main menu, from which students can select one of the four options or go directly to the "Challenge" ("Reto" in Spanish). Figure 3B shows the progress menu where they can see how much they have accomplished.

Figure 4 illustrates the three levels of the micro-learning component, which tackle user performance. Each contains a set of aphorism exercises, which ask the users about their interpretation of aphorisms. The user can choose between two options and evaluate whether their interpretation is correct. The main idea is to make users feel like they compete. First, they can choose between Spanish or English language. Then, the user

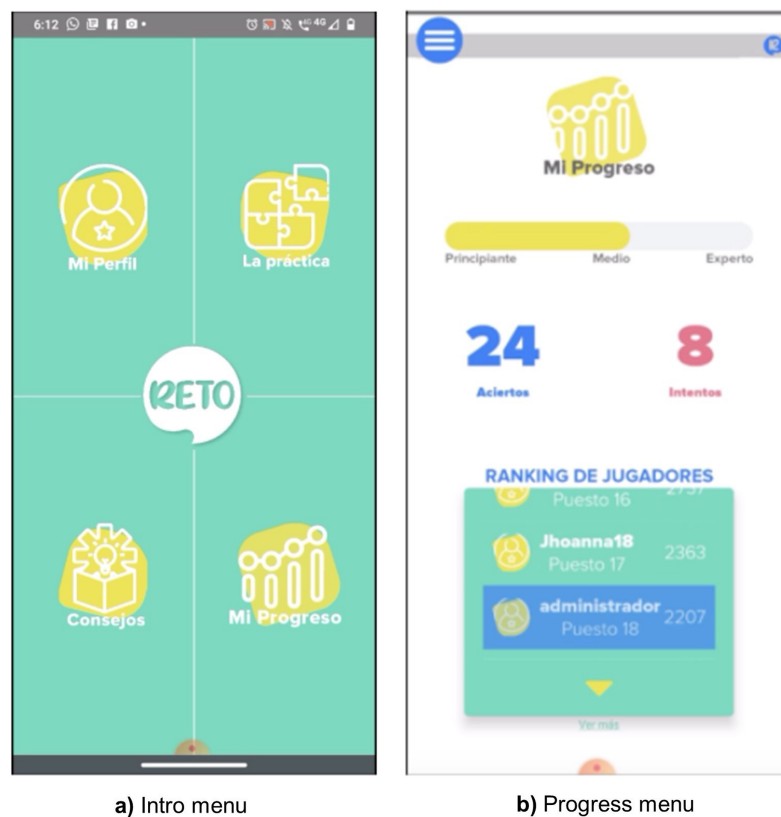

**a)** Intro menu                    **b)** Progress menu

**Figure 3  Screenshots of the EntreLineas application.** (A) Intro menu; (B) progress menu.

begins at the Beginner level with a set of five random aphorisms; if the user finishes the level with the correct interpretations, they can pass to the next level, and the application increases the level of difficulty. In Fig. 5 an example of how the Entrelineas application works is shown. In Fig. 5A, there is an aphorism (in Spanish) with two options in order to choose the closest interpretation. There is also a highlighted word, which is the key for the participant to infer the meaning of the aphorism. In Fig. 5B, participants can observe the feedback provided by the application when they choose the wrong answer.

The game evaluates the user's performance through a time counter and suggestions for the session. Another sub-component is the forum used to comment on any aphorism exercise by the users. Finally, a "Creative Mode" section is unlocked when the user completes all the levels. This sub-component will be described later.

Figure 6 describes a user challenge component, where users create a challenge against another user. The competition begins when the challenge is accepted. Another subcomponent for content creation is "Creative Mode," which allows the user to suggest aphorisms to be added to the application (see Fig. 7). Administrators review these suggestions before publication. The Entrelíneas' components match the four learning phases proposed by the accelerated learning handbook: activation, demonstration, application, and integration (*Ellis, 2001*; *Imel, 2002*). The components mentioned above, such as tips, indicate how the learners encounter the new material in interesting, enjoyable,

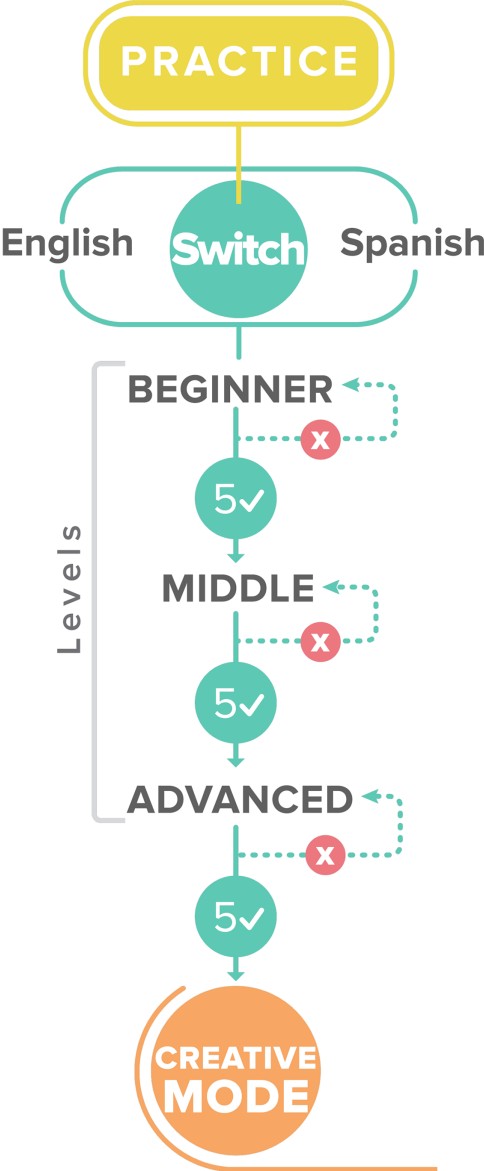

**Figure 4** **Levels of the microlearning component.**

and relevant ways, as in the activation and demonstration stages. The three levels of practice can also allow students to apply what they learned in the tips. These levels allow students to practice and learn from their mistakes, letting them see how new material works in concrete situations and learn to make inferences by doing, supported by gamification. Finally, the challenge and creative modes in which learning is made personal and motivating by discussing with others, proposing content, and obtaining new knowledge, constitute a meaningful stage of learning closer to real-life problems and reflections.
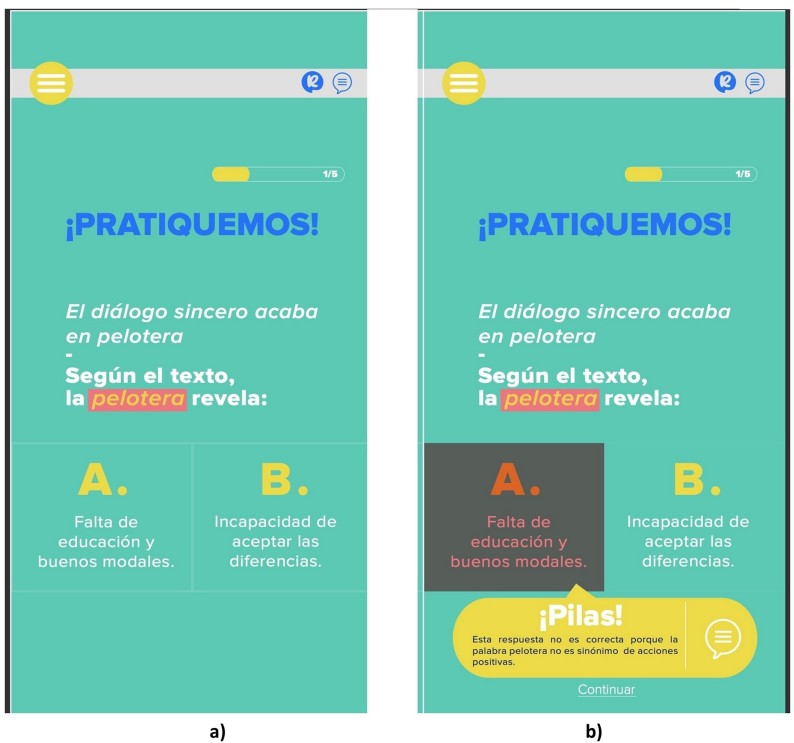

**Figure 5 (A–B) Example of the practice module.**

### Back-end

The application server runs on Amazon Web Services (AWS), which connects to the same provider's Relational Database Server (RDS). Figure 8 shows the network infrastructure deployed on the cloud using Amazon AWS services. The deployment uses Ubuntu 19.04 operating system and an Apache webserver running on an Amazon Elastic Compute Cloud (EC2) instance. The back end runs on Python, where a services library communicates the mobile application and the server. Below, the Back-End modules that integrate with the Front-End are explained. Figure 7 shows the network infrastructure. The mobile application has an authentication module, which uses the OAuth2 standard. It allows users to manage their registration and login/logout actions. When the user logins, the Back-End program sends the client a token authentication to enter the application home screen.

Figure 9 shows the micro-learning module, which catches the users' actions when practicing their reading comprehension skills. First, when the user selects a level and language, the application triggers an API request to retrieve random aphorism exercises from the database, satisfying the requested level and language. Then the user begins to play and answer the questions, and the application updates the results in the database to keep historical learning data.

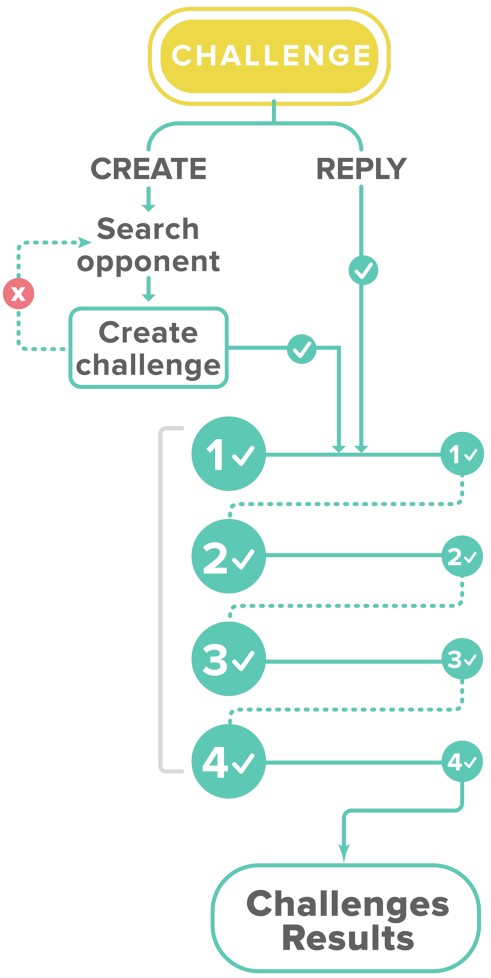

**Figure 6  The flow of the challenge mode.**

# FRAMEWORK EVALUATION METHOD

## Design and sample

The evaluation follows the DBR methodology, which is a flexible and systematized methodology that describes design research as a socially constructed and contextualized process (*Bannan-Ritland, 2003*; *Papavlasopoulou, Giannakos & Jaccheri, 2019*). There are three phases that should be repeated in cycles as many times as needed. First, researchers must analyze and explore the topic of the project, to establish which are the requirements of the desired application. Then, researchers design and build the application using the results from the previous phase. Finally, they evaluate and reflect on the results of the implementation. The previous phases include collaborative work between research faculty and participants (*Cochrane et al., 2017*; *Edelson, 2002*). Some characteristics are intertwined objectives and learning theories, continuous design cycles, enactment, analysis, and redesign (*Wang & Hannafin, 2005*). DBR is suitable for this study since it helps measure the effectiveness of interventions (*Reeves, 2006*) while incorporating applications and understanding how learning occurs using micro-texts. In Fig. 10, the five

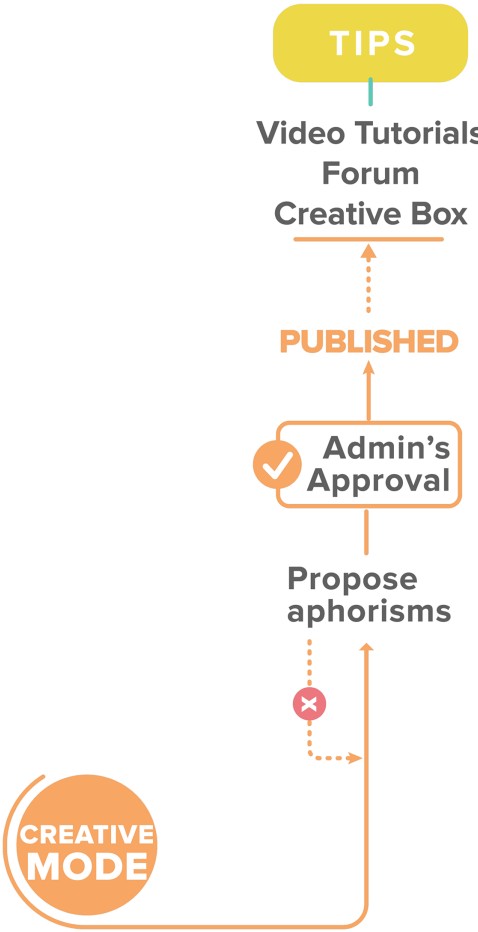

**Figure 7** The flow of the creative mode.

phases applied in this design were adapted from *McKenney & Reeves (2013)*. This figure clarifies how the methodology followed in this article closely follows the DBR definition. Phase 3 of the figure shows an intermediate step of this work, and phase 5 (last column of the figure) gives more details of the methodology implemented in this evaluation.

## Participants

A total of 28 Colombian undergraduate students between 16 to 20 years old participated in the study. The group used the application *Entrelíneas* for 4 weeks. This section explains the process used to evaluate and answer the previously stated research questions.

## Instruments

The study included five phases as part of the DBR strategy. The researchers obtained informed consent forms from each participant. The procedure followed the guidelines of the Internal Ethics Committee of Universidad del Norte, under approval No. 178.

- Analysis and exploration phase: The iPAC Rubric was applied to explore the existing gaps in other reading applications. A pre-test was conducted to diagnose the initial reading performance of the students.

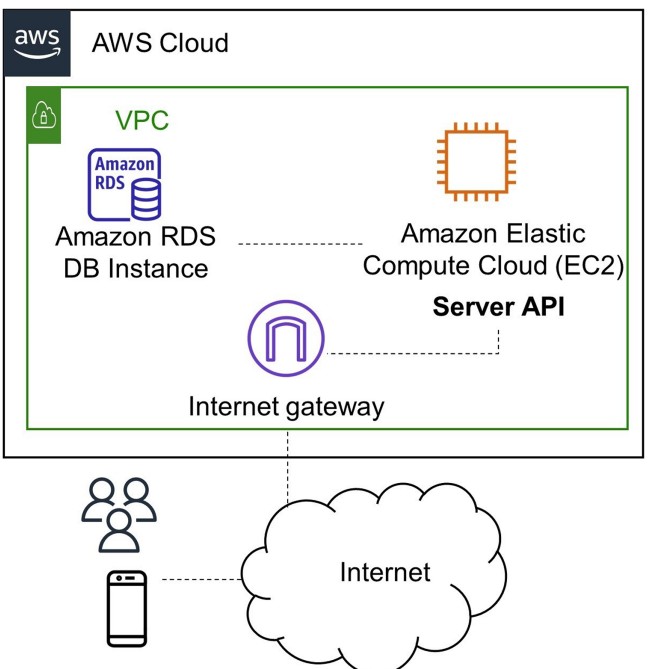

**Figure 8  The network infrastructure of the application's back-end.**

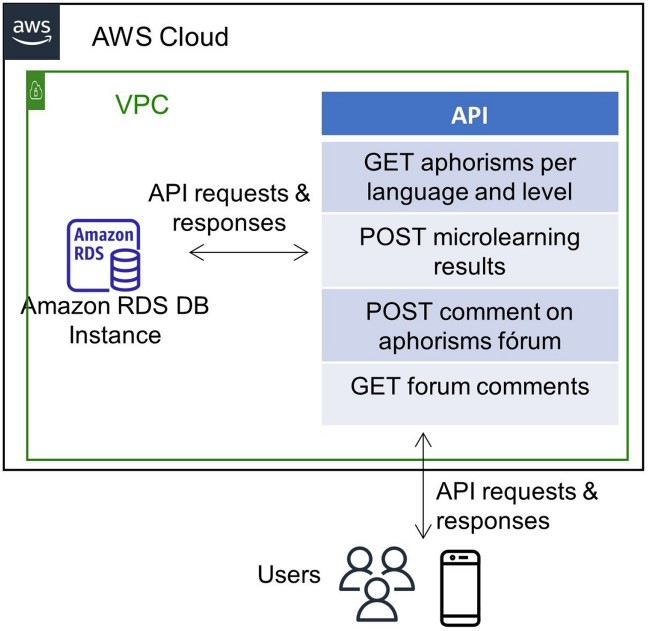

**Figure 9  Creative components.**

- Design and construction of the application phase: The authors prepared a checklist to categorize the aphorisms per level of complexity.
- Evaluation and reflection phase: The first focus group was organized with questions about several technical aspects of the application.

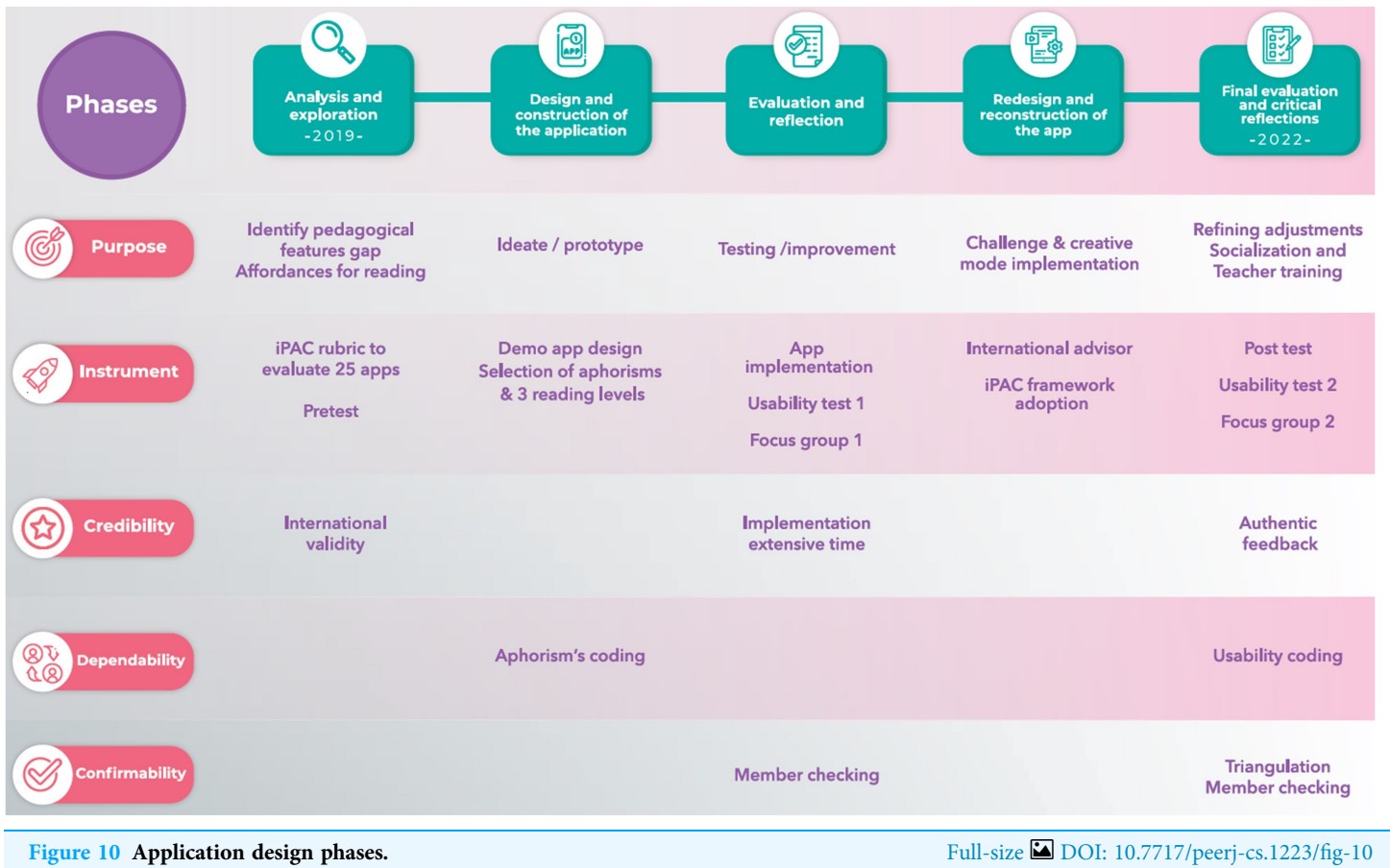

**Figure 10 Application design phases.**

- Redesign and construction of the application phase: Using feedback from students to make the necessary changes for the next development cycle.
- Evaluation and critical reflections phase: A second usability survey was conducted to see students' perceptions. A second session with the focus group allowed the development team to apply the requested changes, and a post-test was conducted to check students' performance after using the application.

## Usability test

The following are the statements the students responded to in the usability test:

1. I think I will use this application when it is available.
2. I find this application unnecessarily complex.
3. I think this application was easy to use.
4. I think that I would need help from a technical person to use the application.
5. I find several functions of the application well integrated.
6. I think there was much inconsistency in this application.
7. I imagine that most people would learn to use this application quickly.

8. I found this application very difficult to use.

9. I felt very confident using this application.

10. I needed to learn many things before I could adequately interact with the application.

### Learning outcomes test

The pre-test and post-test are critical reading components of the nationally standardized exam ("Saber Pro") that the Colombian Institute for the Evaluation of Education (ICFES) facilitates. It has 26 questions that are multiple choice with only one answer, nine are of literal level (34.61%), 10 inferential (38.46%) and seven questions of critical level (26.93%). This test is valid since it contains previously designed questions, and the national education system uses it in Colombia. The questions booklet is found in *Pro (2018)*. The objective of this test was to determine the students' reading comprehension levels before and after the implementation of the app.

### Focus groups

There were two focus groups. Questions related to technological issues:

1. How was your experience with the application in general?

2. What is your opinion about the tutorial's component?

3. What is your perception of the feedback given by the application?

4. What do you think of the interface?

5. Would you download this application to your smartphone?

6. What is your perception of the graphic design of the application?

Questions about pedagogical issues:

1. How was your experience with the application in general?

2. What did you like the most about the application?

3. What was the most challenging thing about the application?

4. What would you like to add to the application?

In the first group, the questions were more about application design. The interface, graphic design, and the second group of questions were adapted from a previous usability study called Heroes, in which researchers incorporated pedagogical features analysis (*Mesurado, Distefano & Falco, 2019*).

## RESULTS

### Usability test

The first research question targets trying the framework by developing a mobile application for reading comprehension using micro-learning. To achieve the application's objective, it offers the aphorisms selection as possible answers to the students corresponding to a short text type. The designed content considered the effectiveness and efficiency of usability aspects. The iPAC framework and the usability categories were used

**Table 3 Application usability assessment indicators.**

| | Usability category | Usability subcategory | Student comment | Analysis |
|---|---|---|---|---|
| Positive indicators | Assessment strategy | Assessment performance | "Another thing that I liked was the competition with my partners." | Students were encouraged to create their own content and participate actively during the application. |
| | Perceived usefulness | Facilitating learning | "To be aware of the answers, compare them with others, and to know when we went wrong." | The more freedom the students enjoy including their ideas in the task, the more engaged they will be. |
| | Perceived usefulness | Improved knowledge | "I liked that we could answer based on what we think." | Students who created new aphorisms had more opportunities to apply critical thinking skills. |
| | Error prevention | Hints and warnings | "It contains videos that guide you before answering and also instructions about doing it." | Students claimed that the videos helped them review the topic when required and reduced the frustration of facing a new topic. |
| | Interactivity feedback and help | Improved communication | "I was sad because no one replied." | Students felt frustrated because their interactions were impossible. |
| | User satisfaction | Expected results | "I felt mad when I made mistakes because I was sent to the beginning, and I had to start over. I set a new purpose, and I kept trying" | The progress bar of the game sometimes stuck and the students felt frustrated and did not want to continue with the game. |
| | Memorability | Question help | "Having two options is not that challenging." | Students suggested more options to make inferences easily. They want to have more aphorisms challenges. |
| Negative indicators | Aesthetics | Color and icons | "Pictures and images must be added." | Students suggested including visual aids to illustrate complex concepts given by the aphorisms content. |

for the final designs. One of the experimental findings was that the students could finish the given tasks more promptly after applying the mentioned framework and usability categories. Therefore, they could have time to explore the challenge mode. Students responding to this feature added that they wanted more questions to compete.

The results obtained from the focus group are displayed in Table 3. The table depicts the application usability assessment indicators according to *Jou et al. (2016)*. The first column shows the negative or positive effects of the indicators according to the comments provided by the students. The second column is the name of the category in general terms. The third column is a more specific subcategory (Assessment performance, Facilitating learning, Improved knowledge, Hints and warnings, Improved communication, Expected results, Question help, and Color and icons). The fourth column shows the students' comments, and the last one is the analysis performed by the study's research team. Students' comments are reflected in the proposed framework by guiding the redesign decisions of the app's final version. For instance, their positive comments about the creation mode, the instructive videos, and the collaborative features confirmed their appropriateness. In the case of negative comments, it helped designers to reduce, delete or modify previous wrong decisions such as the lack of visual aids for aphorisms, expand the progress bar options and avoid delaying students' performance while racing against time or scores.

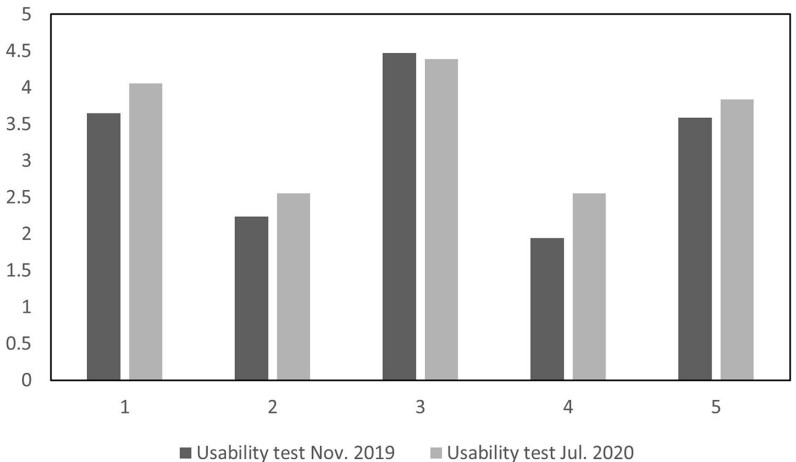

**Figure 11 Usability tests comparison for questions 1 through 5.**

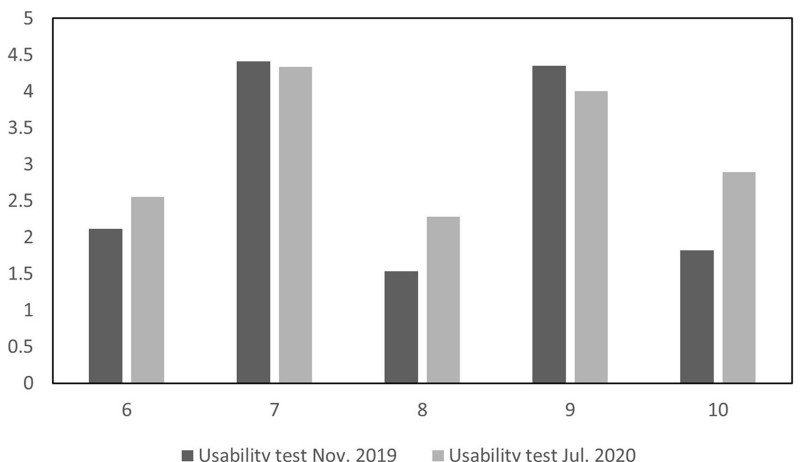

**Figure 12 Usability tests comparison for questions 6 through 10.**

The following are the usability test results, which also respond to the students' perception of the application's usability. The new user interface considered the feedback after the first test, so the users were more satisfied with most test questions with the latest version of the application. In questions 2 and 4, students complained about the application's technical issues and complexity level since the second usability test was applied online. Figure 11 shows the results for questions 1 through 5, while Fig. 12 shows questions 6 through 10.

However, the responses were lower for questions 7 and 9 the second time. Question 7 asked if the users felt most people would learn to use the application quickly. Newly introduced functions may usually decrease the application's usability due to a busier interface that needs to show more options. Question 9 asked about the users' confidence. The results might be related to introducing new possibilities in the interface, which is

**Table 4  Pre and post test scores.**

|  |  | N | Minimum | Maximum | Mean | Standard deviation |
|---|---|---|---|---|---|---|
| Group 1 | Initial score | 14 | 23 | 96 | 63.57 | 24.883 |
|  | Final score | 14 | 23 | 100 | 66.00 | 26.315 |
| Group 2 | Initial percentage | 14 | 35 | 73 | 44.21 | 10.467 |
|  | Final percentage | 14 | 38 | 96 | 61.64 | 23.190 |

expected in an application development process when comparing a first to a second version.

The third research question relates to the students' performance after using the application. It was approached *via* the following findings from the pre and post-test, administered 2 months apart. There is a difference in the results in almost all the questions, but there could be multiple reasons for better post-test results. In this case, the students did not use any other tool to improve their reading comprehension since they only depended on the one the school provided. Also, the time between pre-and post-test might not be enough to impact their capabilities. As part of the evidence of the app's effectiveness, it is outlined the following aspects: There was a 17.31% improvement in the overall performance of the students who used the app for 4 weeks. Improvements were observed in the post-test at the inferential level (8.24%) once the application was implemented in its demo version. Concerning the critical reading level, where students faced seven questions, progressive changes were observed in the correct answers of the post-test. Apart from working on inferences, the aphorisms also encourage the student to take a position on the text. Eventually, if the pedagogical strategy of the short text as a vehicle to increase reading comprehension directs the students to grasp implicit meanings, they could judge the text's structure and evaluate what they are reading. In focus groups, students also indicated how the text's presentation from the perspective of gamification responds to the demands of the context of the student of this generation: the immediacy of digital communication. "The current era is governed by immediacy and instantaneousness, two factors practically dominating human communication" (*Puertas Hidalgo, Cadme & Álvarez Nobell, 2015*). *Entrelíneas* offers integrating the texts with instructional videos, tips, and challenges that allow the user to individualize the learning process and insert their reading habits into their cyberspace where academics, likes, and knowledge in general coexist. These initial results were considered slightly positive after the implementation; see the information in Table 4.

## Pre and post-test scores

The results revealed no significant differences in learning outcomes. The student's average score went from 63.6 to 66 points in the post-test, with equal dispersion. In contrast, the second group's average went from 44 to 61.6, thus making a slight difference and average dispersion. The pre-test and post-test application scores were compared to determine the differences between the groups' performances. The normality analysis was conducted for each period using the SPSS package and the Kolmogorov–Smirnov test. Table 5 summarizes the test results in which the statistical normality is shown for each test.

**Table 5 Kolmogorov–Smirnov test.**

|  |  | Initial score | Final score |
|---|---|---|---|
|  | N | 28 | 28 |
| Normal parameters | Mean | 53.89 | 63.82 |
|  | Standard deviation | 21.167 | 24.439 |
| Maximum extreme differences | Absolute | 0.145 | 0.188 |
|  | Positive | 0.145 | 0.166 |
|  | Negative | −0.115 | −0.188 |
| Test statistic |  | 0.145 | 0.188 |
| Asymptotic sig. (bilateral) |  | 0.135 | 0.013 |

**Table 6 Wilcoxon signed rank test.**

|  | Group 1 | Group 2 |
|---|---|---|
| Z | −0.589 | −2.764 |
| Asymptotic sig (bilateral) | 0.556 | 0.006 |

The pre-test scores show a normal distribution with a significance greater than 0.05 ($p > 0.05$). Nevertheless, the post-test scores are not the same since they do not show a normal distribution ($p > 0.05$). In these cases, it is necessary to apply a t-test for paired means difference, non-parametric, or the Wilcoxon test used to compare two samples of data from the same group of individuals, as is shown in Table 6

The scores of both tests regarding group 1 were compared, and no significant differences were observed (at a significance level of 0.05; $p > 0.05$). This result confirms the observation in the means analysis performed earlier, where the mean scores of both measurements were 63.6 and 66, respectively. A different situation occurred with both tests' scores in group 2 since there was a statistical difference between both measurements (at a significance level of 0.05; $p < 0.05$). This result confirms the difference of almost 17 points observed in the previous means analysis. On the other hand, to determine differences between the proportion of correct answers in the 26 items applied before and after using the tool, a t-test for paired samples was applied at the 95% confidence level. First, as recommended earlier, applying a normality test to each variable is necessary as it is a fundamental requirement for using the test. Table 5 presents the results of the KS (Kolmogorov–Smirnov) normality test available in the SPSS package. The test confirms the normality of the correct answers from the pre-test and post-test applications, with a significance greater than 0.05 ($p > 0.05$). In this sense, the results fulfil the fundamental requirement for applying the t-test for the difference of paired means.

## DISCUSSION

This study aimed to design a general framework for creating micro-learning mobile applications to improve reading comprehension to bridge the gap between students striving to achieve higher reading levels in English and Spanish.

Different abstract software characteristics were listed to respond to each of the necessary features to answer the first research question. The software components that represent those characteristics include creating micro-learning activities, discussion and comments, user challenges, content co-creation, and micro-learning activities for the students. The framework and its components respond to the features needed in such applications. The results were promising for the second question and showed that students' perceptions were generally favourable. After the interaction with the application, students showed interest in certain usability aspects, showing a positive reaction for five categories, and four were negative, as shown in Table 3. A promising follow-up study should be more extensive with multiple tools to compare the application's effective outcome. The constant interaction of the participant students, thanks to the DBR methodology, helped improve the usability features evaluated in the application. Besides, using the iPAC framework allowed the students to participate in the learning process by creating content for the application, thus reflected also in their perception of usability.

For the third research question, the potential of the application can be shown in progress from pre and post-tests. Although it is not expected to observe a rigorous advancement every time a student uses the application, the engagement to continue practising might increase thanks to *Entrelíneas*. The work in *Ciampa (2014)*, *Stephen (2020)* shows that students are more prone to working digitally. Students were motivated during all the application modules, especially the "challenging" module. The application emphasizes gamification and the exercises' speed by giving only two options to select from while answering the questions. Regarding the last question, it is concluded that the blend of the DBR and iPAC allowed a re-evaluation of the missing aspects of the pedagogical components that are usually neglected when designing educational applications. The constant iterations of DBR contributed to improving the application based on the student's needs.

DBR and iPAC evolved the application's structure toward a highly engaging tool based on gamification. Challenge, competition, and cooperation tasks can enhance motivation (*Glover, 2013*; *Jayalath & Esichaikul, 2020*; *Su & Cheng, 2015*). The work in *Jong et al. (2021)* used DBR to show the importance of gamified mobile learning and verbal pedagogical, administrative, and technical intervention in higher education. Furthermore, reducing the multiple answers format was a manner to maintain students' attention while presenting a complex reading strategy, such as making inferences. Another aspect that emerged from the focus groups in terms of usability was the incorporation of visual aids. Students suggested that adding pictures to each aphorism could help them contextualize the exercise quickly. Simultaneously, it would match the gamification purpose since images could promote active participation.

Most of the application implementation for students and teacher training sessions were monitored remotely since the international expert assigned was out of the country, the students had time constraints due to final exams, and the teacher's schedules were tight. In the future, it is expected to be covered face-to-face to collect extra data and implement *Entrelíneas* in other contexts with the exact needs.

## CONCLUSIONS AND FUTURE WORK

DBR is a helpful methodology for building mobile applications for research purposes, and as shown in this work, especially useful for constructing a technological solution to reading comprehension. Several versions of the framework and the mobile application prototype were designed to test with users. Another study (*Wong et al., 2011*) used what the authors called a micro-cycle (*Leinonen et al., 2016*) to design several learning applications. The literature and this work's findings show the advantage of using the DBR methodology for similar projects.

This work used five phases to evaluate how effective it was to develop a framework design for a reading comprehension micro-learning mobile application. Some of the implications of this DBR are related to credibility, confirmability, and dependability applications. All the above are relevant in obtaining authentic results, triangulating the instruments' information, and accurately organizing the qualitative feedback accurately (*Pool & Laubscher, 2016*).

This study makes a critical contribution since researchers could use the framework for developing similar applications, which is the framework's original purpose. The components design allows scholars to select only the functions the new application requires or design depending on the project's time and resources. It also enables a smooth implementation using cloud computing microservices.

### Funding

This work was supported by the Ministry of Sciences of Colombia (MinCiencias). The funders had no role in study design, data collection and analysis, decision to publish, or preparation of the manuscript.

### Grant Disclosures

The following grant information was disclosed by the authors:
Ministry of Sciences of Colombia (MinCiencias).

### Competing Interests

The authors declare that they have no competing interests.

### Author Contributions

- Heydy Robles conceived and designed the experiments, analyzed the data, authored or reviewed drafts of the article, and approved the final draft.
- Miguel Jimeno conceived and designed the experiments, analyzed the data, prepared figures and/or tables, authored or reviewed drafts of the article, and approved the final draft.
- Karen Villalba conceived and designed the experiments, analyzed the data, authored or reviewed drafts of the article, and approved the final draft.

- Ivan Mardini performed the experiments, performed the computation work, prepared figures and/or tables, and approved the final draft.
- César Viloria-Nuñez analyzed the data, prepared figures and/or tables, and approved the final draft.
- Wendy Florian performed the experiments, performed the computation work, prepared figures and/or tables, and approved the final draft.

## Ethics

The following information was supplied relating to ethical approvals (*i.e.*, approving body and any reference numbers):

The internal Ethics Committee of Universidad del Norte approved the study (Approval no. 178).

## Data Availability

The code of the mobile application is available in the Supplemental File.

## Supplemental Information

Supplemental information for this article can be found online at http://dx.doi.org/10.7717/peerj-cs.1223#supplemental-information.

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
