# Peer review of "Design of a micro-learning framework and mobile application using design-based research"

_PeerJ Computer Science, doi:10.7717/peerj-cs.1223_

## Round 0.1 · original submission · Major Revisions

· Academic Editor

Major Revisions

The paper is interesting, but it needs more improvements in terms of experiments setup, methodology and results.

It would be great if the authors include a chart or figure that clarifies the experiments methodology and results.

Also the authors should work on the comments from the reviewers.

·

Basic reporting

- The authors should use pasive voice instead of active voice in multiple occasions on the paper
- The authors should use present simple or present perfect instead of simple past tense on multiple occasions in the paper
- The paper has some misspellings
- There are some issues in the table and figure references
- I have noticed that there are a lot of short sentences that don't make sense. The authors should consider making the sentences more extended and more meaningful.
- The authors have provided sufficient references and background regarding their subject
- Check the attached file for some examples of the above issues

Experimental design

- The authors have defined their research questions and answered them clearly in the results section
- In figure 1 (b), are the feedback provided by the students? If yes, then I think you should split generating learning statistics from the student’s entity as it should be generated by the system or another entity
- Line 208: You should append “On the right side, we observe the feedback provided by the application when the participant chooses the wrong answer.” to the previous paragraph instead of creating a paragraph of just one sentence.
- Figure 5 should be split into (a) and (b) and referred to by using them instead of “left” and “right”.
- Personally, I think providing the screenshots in English would be better than just explaining them in English. Nevertheless, this is just an opinion.
- Check the attached file for further remarks

Validity of the findings

- The authors have compared and analyzed the results of the comparisons and provided a sufficient discussion of how their work impacted the participants of the experiment
- The data is valid
- The conclusion and future work is suitable
- Check the attached file for further remarks

·

Basic reporting

- Are the authors and their affiliations properly formatted?
Ans: yes
- Is the title of the paper appropriate?
Ans: yes
- Is the abstract well written? If not, might the author(s) need to rewrite it and focus on novelty and comparative analysis performed in the paper? (The abstract should explain what the problem is and how it is solved here).
Ans: Yes, to some extent
- Are the keywords used acceptable? The authors suggested using the keywords by selecting more relevant terms.
Ans: Yes, to some extent

Experimental design

Does the description of technical details of the methodologies and the proposed need to be added, and in which sections?
Ans: Yes in the conclusions and results


- Is the computational burden of the algorithm/proposed model/method discussed? Does need to explain more about the used approach and methodology? Is there any flowchart, diagram, etc. needed to be added or modified?
Ans: Yes, to some extent



- Is the practical contribution well challenged enough? (Ex: the study may need to compare to other approaches, the advantages and disadvantages should be compared to the existing methods need to be presented, etc.).
Ans: Yes, to some extent

- Is/are the dataset(s) well explained like what is the size of the picture, resolution, colour of masks, features, classes, training, testing, etc clearly to the reader (may need to explain the dataset(s) in the table(s)).
Ans: Yes, to some extent

Validity of the findings

Are the experiments and quantitative results presented well? What do you suggest? The results section must contain quantitative results in the form of a table and these must be discussed in detail to convey to the reader how these results are obtained and why they are valuable.
Ans: Yes, to some extent


- This section of experiments should be made strong. This is the backbone of the study. Please add your comment you think could improve this section.
Ans: Yes, to some extent


- Is there any need to include the description of the metrics used in their experiments, their mathematical formula etc? Ex: accuracy, precision, recall and f-measure. Or, include a separate table to show these metrics.
Ans: No
- Are the metrics used in the experiment enough?
Ans:Yes, to some extent


- Is/are the figure(s)/image(s) enough to prove the results? Does the author(s) need to provide1/2 or more tables to substantiate and clearly evaluate the study?
Ans: Yes, to some extent

- Is/are the table(s) enough to prove the results? Does the author(s) need to provide1/2 or more figures to substantiate and clearly evaluate the study?
Ans : Yes, to some extent
- Are the figures and tables given in the result part well discussed? Which figure(s)/table(s) do you think needs to be discussed or not well discussed?
Ans: Yes, to some extent
- Is there any figure(s) is/are not clear, axes missed, quality needs to be improved, etc? Authors should not use any screenshots (snapshots) for Figures and or Tables
Ans: No

- Are Image, Figures and Tables aligned with text?
Ans: Yes, to some extent

- Are Images, Figures, Tables, Captions, etc. formatted well?
Ans: Yes, to some extent
- Are the equations well formatted and/or numbered properly? Please specify which equations need to be checked.
Ans: Yes, to some extent
- Is there any extra spaces needed to be removed? Ex: between, paragraphs/figures/tables/sections, at the bottom/top, etc.
Ans: No

Additional comments

Are the experiments and quantitative results presented well? What do you suggest? The results section must contain quantitative results in the form of a table and these must be discussed in detail to convey to the reader how these results are obtained and why they are valuable.
Ans: Yes, to some extent


- This section of experiments should be made strong. This is the backbone of the study. Please add your comment you think could improve this section.
Ans: Yes, to some extent


- Is there any need to include the description of the metrics used in their experiments, their mathematical formula etc? Ex: accuracy, precision, recall and f-measure. Or, include a separate table to show these metrics.
Ans: No
- Are the metrics used in the experiment enough?
Ans:Yes, to some extent


- Is/are the figure(s)/image(s) enough to prove the results? Does the author(s) need to provide1/2 or more tables to substantiate and clearly evaluate the study?
Ans: Yes, to some extent

- Is/are the table(s) enough to prove the results? Does the author(s) need to provide1/2 or more figures to substantiate and clearly evaluate the study?
Ans : Yes, to some extent
- Are the figures and tables given in the result part well discussed? Which figure(s)/table(s) do you think needs to be discussed or not well discussed?
Ans: Yes, to some extent
- Is there any figure(s) is/are not clear, axes missed, quality needs to be improved, etc? Authors should not use any screenshots (snapshots) for Figures and or Tables
Ans: No

- Are Image, Figures and Tables aligned with text?
Ans: Yes, to some extent

- Are Images, Figures, Tables, Captions, etc. formatted well?
Ans: Yes, to some extent
- Are the equations well formatted and/or numbered properly? Please specify which equations need to be checked.
Ans: Yes, to some extent
- Is there any extra spaces needed to be removed? Ex: between, paragraphs/figures/tables/sections, at the bottom/top, etc.
Ans: No

---

## Round 0.2 · Major Revisions

· Academic Editor

Major Revisions

- The authors should discuss the weaknesses and limitations of the current studies such as Competency-Based Education (CBE) (79), 22, 12 in Related Work Section.
- The authors should describe the gap point in the final paragraph of the Related Work Section.
- The authors should include further studies in this area. 3 studies aren't sufficient.
- The author should explain in few lines how they measured the usability of the current framework. Also they should include the attributes of usability of the current framework focused in this paper.
- Did the authors consider these attributes of usability "The included attributes are A) effectiveness, B) efficiency, C) satisfaction, D) productivity, E) universality, F) learnability, G) appropriateness, H) recognizability, I) accessibility, J) operability, K) aesthetics, and L) error protection" into this paper?
- ISO 25010 model considers all attributes of usability except to D) productivity, E) universality and L) error protection. So did the authors consider these three attributes in their framework?

- I think the research questions should be changed from what to how. Please rephrase these questions

- How did students reflect their comments to the proposed framework?

- "Cloud Framework" is only mentioned in Section 4, but it's not mentioned in the Abstract or even the paper article.

·

Basic reporting

- The authors have reflected the comments of the review on the document. It is now suitable.

Experimental design

- The authors have reflected the comments of the review on the document. It is now suitable.

Validity of the findings

- The authors have reflected the comments of the review on the document. It is now suitable.

---

## Round 0.3 · accepted · Accept

· Academic Editor

Accept

The authors addressed my concerns in the revised paper.